# Disrupted Pallido-Thalamo-Cortical Functional Connectivity in Chronic Disorders of Consciousness

**DOI:** 10.3390/brainsci11030356

**Published:** 2021-03-11

**Authors:** Anna Sontheimer, Bénédicte Pontier, Béatrice Claise, Carine Chassain, Jérôme Coste, Jean-Jacques Lemaire

**Affiliations:** 1Institut Pascal, Université Clermont Auvergne, Centre National de la Recherche Scientifique, Institut National Polytechnique-Clermont, Centre Hospitalier Universitaire de Clermont-Ferrand, F-63000 Clermont-Ferrand, France; bpontier@chu-clermontferrand.fr (B.P.); cchassain@chu-clermontferrand.fr (C.C.); jcoste@chu-clermontferrand.fr (J.C.); jjlemaire@chu-clermontferrand.fr (J.-J.L.); 2Service de Radiologie, Unité de Neuroradiologie, Centre Hospitalier Universitaire de Clermont-Ferrand, F-63000 Clermont-Ferrand, France; bclaise@chu-clermontferrand.fr

**Keywords:** cerebral networks, disorders of consciousness, functional resting-state connectivity, mesocircuit model, pallidum, thalamus

## Abstract

Chronic disorders of consciousness (DOC) encompass unresponsive wakefulness syndrome and minimally conscious state. Their anatomo-functional correlates are not clearly defined yet, although impairments of functional cortical networks have been reported, as well as the implication of the thalamus and deep brain structures. However, the pallidal functional connectivity with the thalamus and the cortical networks has not been studied so far. Using resting-state functional MRI, we conducted a functional connectivity study between the pallidum, the thalamus and the cortical networks in 13 patients with chronic DOC and 19 healthy subjects. We observed in chronic DOC patients that the thalami were no longer connected to the cortical networks, nor to the pallidums. Concerning the functional connectivity of pallidums, we reported an abolition of the negative correlation with the default mode network, and of the positive correlation with the salience network. The disrupted functional connectivity observed in chronic DOC patients between subcortical structures and cortical networks could be related to the mesocircuit model. A better understanding of the DOC underlying physiopathology could provide food for thought for future therapeutic proposals.

## 1. Introduction

Chronic disorders of consciousness (DOCs) following a coma, are mostly encountered after traumatic head injury, stroke or cardiorespiratory arrest [1,2]. The clinical pictures are the vegetative state or unresponsive wakefulness syndrome (UWS) and the minimally conscious states (MCS), which are respectively characterized by the recovery of sleep/wake cycles without conscious behaviors [3], and by inconsistent reproducible conscious behaviors [4]. Their related anatomo-functional correlates are however still not mastered, even though impairments of functional cortical networks have been reported. Indeed, the functional connectivity and the metabolic activity are decreased within the default mode network (DMN) [5,6]. The executive control (CEN), salience (SN), sensorimotor, auditory and visual networks are also significantly impaired (see [7] for review).

The role of the deep brain structures has received less interest, but substantial data arise. In that respect, the mesocircuit model enables us to interpret the impact of widespread lesions within the cortico-striato-pallido-thalamo-cortical circuitry that downregulate the cortical activity. According to this model, the activity decreases within thalamo-cortico-striatal loops following neuronal loss in the central thalamus, which leads to the deafferentation of the striatum, and consequently to a loss of inhibition of the internal pallidum. This disinhibition leads to active inhibition of the thalamus, and is proposed to have a causative role in DOC [8,9]. In DOC patients, structural connections between deep gray nuclei, notably the pallidum and the thalamus, and frontal cortices decrease [10], while the lesion burden in the central gray nuclei seems correlated with the severity of recovery scores [11]. The disruption of DMN thalamo-cortical functional connectivity was observed in experimental dexmedetomidine loss of consciousness [12] and in DOC patients [13]. In the same vein electric stimulation of the thalamus, the tegmentum and the pallidum, either experimental or clinic, have illustrated their roles in consciousness processes [14,15,16,17,18]. In all of these cases, the dysfunctions of cortico-subcortico-cortical loops were interpreted through the prism of the well-known sensori-motor, associative and limbic loops (e.g., [19]). Among the central gray nuclei, the pallidum has received little attention, and notably the specific study of its functional connectivity with cortices has not yet been studied in DOC patients. Hypothesizing that specific functional features could be observed, we conducted a clinical study exploring the functional connectivity between the thalamus, the pallidum and cortical networks in chronic DOC patients using resting-state functional MRI (fMRI).

## 2. Materials and Methods

### 2.1. Participants

Twenty healthy adult volunteers, with no history of neurological or psychiatric disorders, participated as a control group. Data from one participant had to be discarded due to discomfort during the scanning, prompting to premature termination of the experiment. The final dataset included 19 healthy subjects (10 females; 16 right-handed, 2 left-handed and 1 ambidextrous; mean age = 27.9 ± 9.6 years, range 21–56 years).

Fifteen patients with DOC participated in the study. Two patients were excluded due to movements in the MRI machine. At the end, 13 patients were enrolled in the functional connectivity study: 4 UWS, 9 MCS; 6 females; 12 right-handed, 1 left-handed; mean age = 38.5 ± 14.3 years, range 20–63 years; mean duration of DOC = 4.3 ± 3.2 years, range 10 months-12 years and 7 months. The diagnosis was assessed by considering the highest score on the Coma Recovery Scale-Revised (CRS-R, [20,21]), out of a minimum of five assessments in a short period of time, as recommended to reduce misdiagnosis [22]. The CRS-R was routinely administered prior to MRI, in order to estimate the consciousness state at the time of the MRI assessment. Due to the fluctuating nature of consciousness in patients with DOC, these CRS-R scores may differ from the diagnosis. Patients’ characteristics and CRS-R scores at the time of MRI are shown in Table 1. Patients P1 to P5 participated in an already published pallido-thalamic stimulation study [14].

### 2.2. Data Acquisition and Preprocessing

Images were acquired using a 3T MRI scanner (GE, Discovery MR750, General Electric, Boston, MA, USA) with a 32-channel head coil. High-resolution T1-weighted structural images were obtained with a three-dimensional inversion recovery gradient-echo sequence (BRAVO), yielding 288 interleaved slices of 1.4-mm thickness in the axial plane (TR = 8.8 s, TE = 3.6 ms, TI = 400 ms, flip angle = 12, Field of View (FOV) = 240 mm; resulting voxel size = 0.47 × 0.47 × 0.7 mm^3^). Resting-state functional MRI acquisition was performed using a whole-brain gradient-echo EPI sequence (interleaved acquisition resulting in 48 contiguous axial slices of 4-mm thickness: TR = 3 s, TE = 30 ms, flip angle = 90, FOV = 240 mm; resulting voxel size = 3.75 × 3.75 × 4 mm^3^). One hundred and thirty-five resting-state functional volumes were acquired. As it was not possible to check the opening or closing of the eyes during the acquisition and considering the uncooperative condition of consciousness disorders, no instructions were given and a black screen was displayed to patients and healthy subjects.

Data preprocessing was performed using SPM12 (Wellcome Trust Centre for Neuroimaging) implemented in MATLAB (version R2016a, MathWorks, Natick, MA, USA). The first five volumes were discarded as dummy scans. Images were slice-time corrected with reference to the acquisition time of the middle slice, and motion corrected with realignment to the first volume. Outliers were detected using ART (http://www.nitrc.org/projects/artifact_detect, accessed on 6 September 2016) and defined as volumes with realignment parameters >2 mm and 2 degrees, or with signal intensity changes >4 times the standard deviations. The T1 structural volume was co-registered and segmented into gray matter, white matter and cerebrospinal fluid. Automatic segmentation of gray matter voxels part of the thalamus and pallidum is not efficient with SPM12 because they are considered as part of the white matter on the implemented tissue probability maps (TPMs) [23]. This is because the TPMs have been developed from T1-weighted sequences on which the lateral part of the thalamus and the entire pallidum appear with an intensity close to the white matter. These structures being regions of interest for our study, a new TPM has been in-house-developed to ensure that the corresponding voxels are not classified as white matter and are not excluded from the functional connectivity analysis (Figure 1). The in-house-developed TPM was created by overlapping the SPM12 TPM volumes and the pallidums and thalami volumes with artificial values settled to 1 for the gray matter probability map and 0 for the white matter probability map. Pallidums and thalami volumes were defined from the Neuromorphometrics Atlas implemented in SPM12 (provided by Neuromorphometrics, Inc. under academic subscription). The overlap was generated with MRIcron (from http://www.nitrc.org/projetcs/mricron, accessed on 6 February 2019), the SPM12 TPM and the pallidums and thalami volumes being in the same Montreal Neurological Institute (MNI) space, the resulting in-house-developed TPM was also in the MNI space.

The quantification of cerebral lesions in patients was carried out by measuring the numbers of voxels belonging to the cerebral networks, thalami and pallidums (Appendix A). Patient P1 had no more left pallidum and has been excluded from the left pallidal functional connectivity analysis. Due to extensive lesions altering anatomical data processing, the spatial normalization into the MNI space was estimated from the functional data using an EPI template, with the SPM12 default values (affine regularization with ICBM/MNI space template; nonlinear frequency cutoff: 25; nonlinear iterations: 16; nonlinear regularization: 1). Images were resampled at 2 × 2 × 2 mm^3^ using a 4th degree B-Spline interpolation. The anatomical data were spatially normalized using the same deformation parameters. The normalized volumes were visually checked with the implemented quality assurance tool, based on slices display with MNI boundaries, to prevent abnormal results.

Since the signal of interest is considered to belong to the gray matter, the white matter, cerebrospinal fluid, realignment and scrubbing parameters were used as confounders for nuisance regression, with linear detrending. A band-pass filter (0.008 Hz–0.09 Hz) was applied.

### 2.3. Analysis of Functional Connectivity

Resting-state functional connectivity analysis was performed using the CONN functional connectivity toolbox (v.19.c, http://www.nitrc.org/projects/conn, accessed on 1 July 2020, [24]). A region of interest (ROI)-to-ROI analysis was conducted, by computing the Fisher-transformed bivariate correlation coefficients between time-series of each pair of ROIs. Subcortical ROIs of the right and left thalamus and pallidum were defined according to the Harvard-Oxford subcortical atlas implemented in CONN. Cortical ROIs were defined according to the CONN network cortical ROIs atlas derived from independent component analysis of Human Connectome Project dataset (497 subjects), consisting of 8 networks including 32 ROIs: DMN (4 ROIs, medial prefrontal cortex, precuneus/retrosplenial cortex, right and left lateral parietal cortex); SN including ventral attention network (7 ROIs, anterior cingulate cortex, right and left anterior insula, right and left supramarginal gyrus, right and left rostral prefrontal cortex); CEN (4 ROIs, right and left lateral prefrontal cortex, right and left posterior parietal cortex); dorsal attention network (4 ROIs, right and left intraparietal sulcus, right and left rostral prefrontal cortex); language network (4 ROIs, right and left posterior superior temporal gyrus, right and left inferior frontal gyrus); visual network (4 ROIs, medial visual cortex, occipital visual cortex, right and left visual cortex); sensorimotor network (3 ROIs, right and left lateral sensorimotor cortex, superior sensorimotor cortex); and cerebellar network (2 ROIs, anterior and posterior). Functional connectivity between the subcortical ROIs and the cortical ROIs was assessed, considering significant connection values for *p*-FDR corrected < 0.05, and significant cluster for ROI-level *p*-FDR corrected < 0.05 (MVPA omnibus test). In second-level analysis, a paired *t*-test was performed between controls and patients (controls > patients, *p*-FDR corrected < 0.05).

## 3. Results

### 3.1. Thalamo-Cortical Functional Connectivity

In the control group, the right and left thalamus were functionally connected to each other, and showed different connectivity patterns. The activity of the right thalamus was correlated to the right pallidum, SN (6 ROIs, anterior cingulate cortex, right and left anterior insula, right and left supramarginal gyrus, right rostral prefrontal cortex), and the right posterior parietal cortex belonging to CEN. The activity of the left thalamus was correlated to bilateral pallidum, SN (4 ROIs, anterior cingulate cortex, left anterior insula, left supramarginal gyrus, left rostral prefrontal cortex), the left inferior frontal gyrus belonging to the language network, the left lateral prefrontal cortex belonging to CEN and anti-correlated to the right lateral prefrontal and posterior parietal cortices belonging to CEN, the medial visual network and the right lateral sensorimotor network (Figure 2a).

In DOC patients, the right and left thalamus were functionally connected to each other. There was no remaining thalamo-cortical, nor thalamo-pallidal functional connectivity.

The between groups second-level analysis showed significantly greater functional connectivity in the control group compared to patients between the right thalamus, the SN (4 ROIs, anterior cingulate cortex, right anterior insula, right supramarginal gyrus, right rostral prefrontal cortex) and the right posterior parietal cortex belonging to CEN; it also showed significantly greater functional connectivity between the left thalamus, the SN (2 ROIs, anterior cingulate cortex and left rostral prefrontal cortex) and the left lateral prefrontal cortex belonging to CEN (Figure 2b).

### 3.2. Pallido-Cortical Functional Connectivity

In the control group, the right and left pallidum were functionally connected to each other. The activity of the right pallidum was correlated to bilateral thalamus, and SN (6 ROIs, anterior cingulate cortex, right and left anterior insula, right and left supramarginal gyrus, right rostral prefrontal cortex); and anti-correlated to the left lateral parietal cortex belonging to DMN. The activity of the left pallidum was correlated to the left thalamus, the SN (5 ROIs, anterior cingulate cortex, right and left anterior insula, right and left rostral prefrontal cortex) and the anterior cerebellar network; it was also anti-correlated to DMN (4 ROIs, medial prefrontal cortex, precuneus/retrosplenial cortex, right and left lateral parietal cortex) (Figure 3a).

In DOC patients, the right and left pallidum were functionally connected to each other. The activity of the right pallidum was correlated to the right posterior part of superior temporal gyrus belonging to the language network. The activity of the left pallidum was anti-correlated to the right rostral prefrontal cortex belonging to SN (Figure 3b).

The between groups second-level analysis showed significantly greater functional connectivity in the control group compared to patients between the right pallidum, the right anterior insula belonging to SN (positive correlation) and the right lateral parietal cortex belonging to DMN (negative correlation); similar results were also found between the left pallidum, the right rostral prefrontal cortex belonging to SN (positive correlation) and DMN regions (precuneus/retrosplenial cortex, right lateral parietal cortex; negative correlation) (Figure 3c).

The statistical values are detailed in Appendix A.

## 4. Discussion

Our aim was to investigate thalamo-cortical and pallido-cortical functional connectivity in chronic DOC patients, while being willing to explore the functional status of cortico-subcortical loops.

In our series, we observed that the thalami dramatically lost their functional connectivity with cortical networks and the pallidum, whereas the pallidum showed very limited functional connectivity with the right language and salience networks, respectively with the superior temporal gyrus (right pallidum) and the rostral prefrontal cortex (left pallidum; anticorrelation). The interhemispheric functional connectivity of thalami and pallidum was preserved. When compared to controls (Figure 2b and Figure 3c), the thalami and the pallidum showed lower functional connectivity features, respectively with the CEN and SN, and with the DMN and SN. The analysis of the control group showed physiological functional relationships between the thalamus and the pallidum within the right and left hemispheres, as well as with cortices. Indeed, the thalami had mainly positive correlations with the SN and CEN, whereas the pallidum balanced positive correlations with the SN and negative correlations with the DMN.

Roughly, these findings reveal physiological cortico-sub-cortical functional connectivity and disruptions in DOC, which all together are supported by the well-known cortico-subcortical loops (e.g., [18]). They are also in favor of the possible implication of little known direct pallido-cortical connectivity, such as through direct pallido-cortical structural projections, particularly with prefrontal cortices and basal forebrain [17,25], and models of cortico-subcortical connectivity that inferred the role of a direct corticopetal connectivity from the pallidum with prefrontal, motor and cingulate cortices [26,27]. Our findings also help to figure out the mechanisms of the pallido-thalamic electric stimulation that lead to the apparition of conscious behaviors in chronic DOC patients, associated with an increase of the metabolism of DMN regions involved in self-awareness [14].

Technically, the fMRI analysis of relationships between cortices and deep brain structures has received less interest than the historical structural methods, however the correlation between thalamic nuclei and the SN and the anticorrelation between the pallidum and DMN cortices were already reported in healthy subjects [28,29,30].

Interestingly we also observed, in healthy subjects, asymmetric interhemispheric functional connectivity, where one thalamus and one pallidum, respectively the left and the right, had bilateral functional connectivity, respectively with the two pallidum and the two thalami. This last finding lifts the veil on the already suggested hemispheric dominance of deep brain structures (see e.g., [31,32]).

The aim of the study was to investigate the thalamo-pallido-cortical functional connectivity in chronic DOC, the design was not built to focus on differences between UWS and MCS patients. No significant difference was found between these subgroups, whatever the ROI-to-ROI connection, probably due to a lack of statistical power induced by the small and inhomogeneous samples sizes (4 UWS, 9 MCS). To overcome this limitation and allow the distinction between UWS and MCS patients, more subjects should be included in further studies. Moreover, a correlation analysis between the functional connectivity results and the structural data, whether in terms of gray matter integrity or white matter structural connectivity, could provide more information on the underlying pathophysiological mechanisms.

## 5. Conclusions

To our knowledge, this is the first study investigating the pallidal functional connectivity in chronic DOC, pointing out specific functional disruptions of the cortico-sub-cortical circuitry that involves the mesocircuit model [7]. Our work also illustrates the interest of the development of improved TPMs for the automatic segmentation of subcortical structures. In this way, our method implemented in SPM12 shows that the topological analysis of canonical brain functions relying on cortical networks [27] is amenable to cortico-sub-cortical exploration, as reported above.

Finally, a better understanding of the DOC underlying physiopathology, involving both cortical and subcortical analysis, could provide food for thought for future therapeutic proposals.

## Figures and Tables

**Figure 1 brainsci-11-00356-f001:**
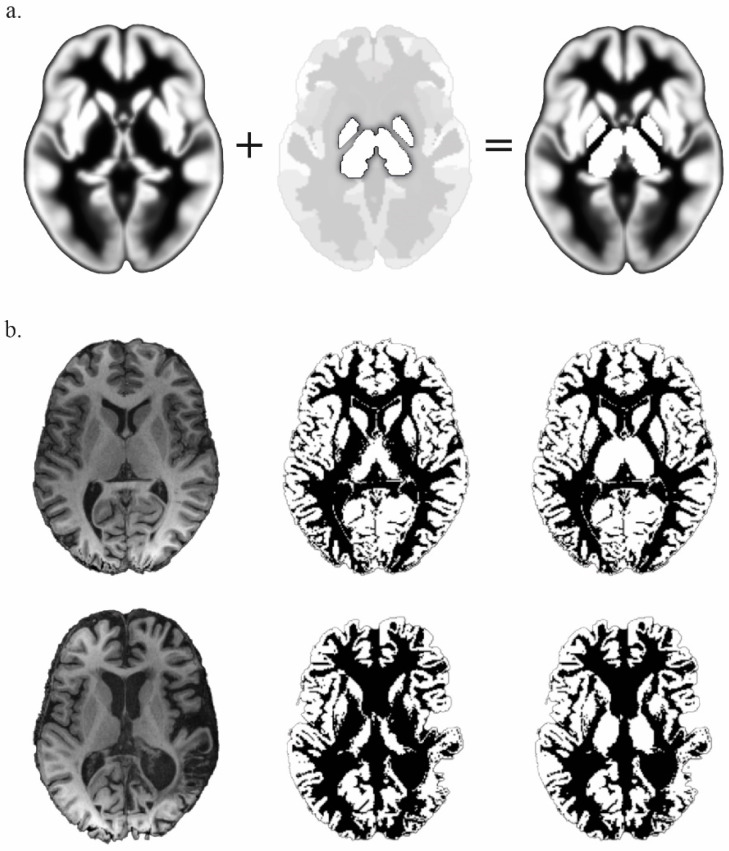
(**a**) In-house-developed gray Tissue Probability Map (TPM) (**right**), elaborated by the overlap of SPM12 gray TPM (**left**) and the pallidums and thalami from the Neuromorphometrics Atlas with a fixed value of 1 (**middle**). (**b**) Comparison of the results of the gray matter segmentation from a normalized-T1 (**left**) with the SPM12 TPM (**middle**) and the in-house-developed TPM (**right**), in a control subject (**top** line) and in a patient (**bottom** line).

**Figure 2 brainsci-11-00356-f002:**
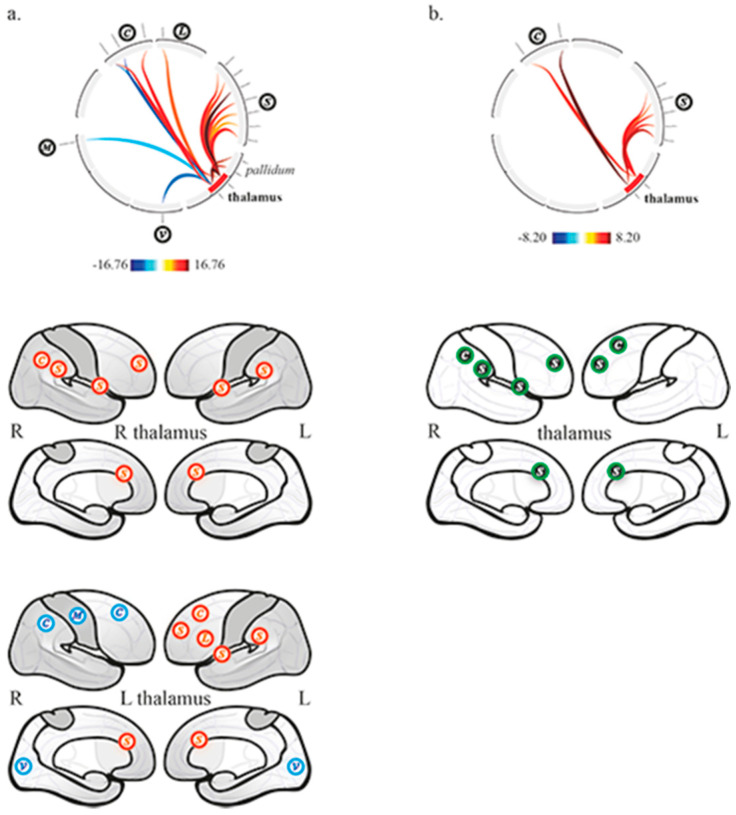
Thalamic functional connectivity (**a**) in the control group, and (**b**) between controls and patients (controls > patients). Top row, circular depiction with color-coded T-scores. Bottom rows, (**a**) display of network related cortices, correlated (red) and anticorrelated (blue); (**b**) display of differences between groups, higher functional connectivity in controls compared to patients (green with black background). C, executive network; L, language network; S, salience network; V, visual network, M, sensorimotor network.

**Figure 3 brainsci-11-00356-f003:**
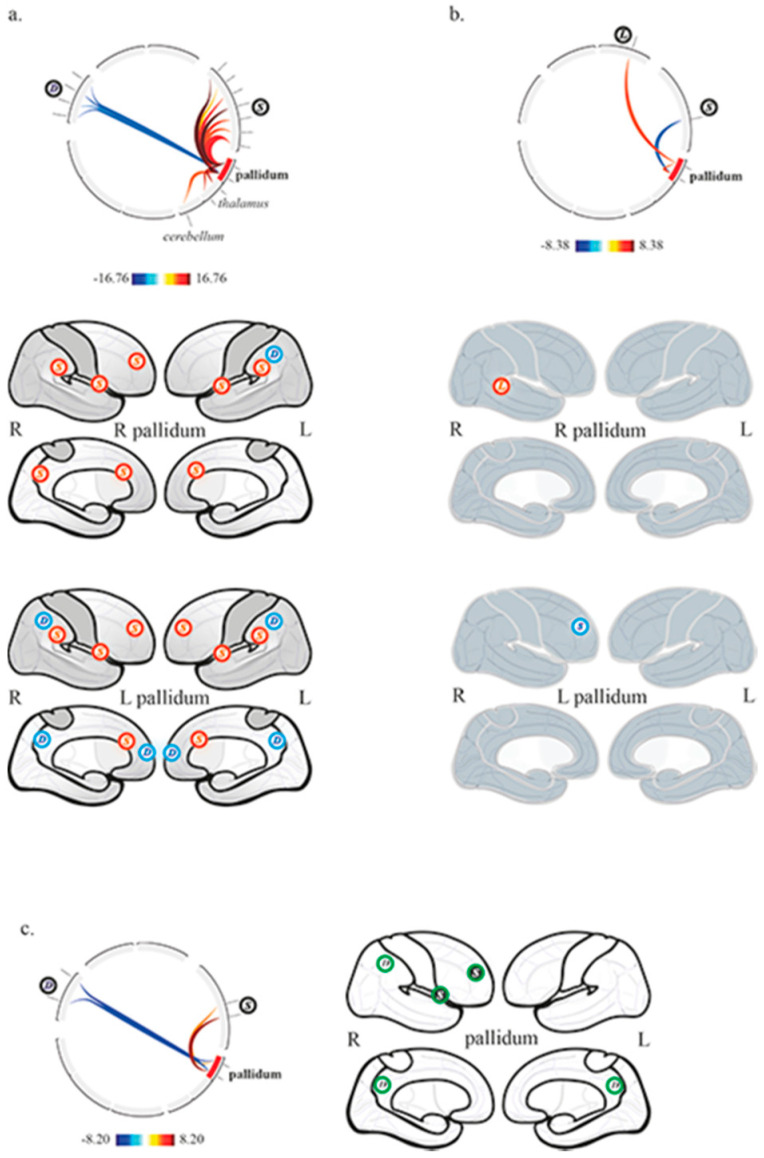
Pallidal functional connectivity (**a**) in the control group, (**b**) in patients with chronic disorders of consciousness, and (**c**) between controls and patients (controls > patients). Top row, circular depiction with color-coded T-scores. Bottom rows, (**a**,**b**), display of network related cortices, correlated (red) and anticorrelated (blue); (**c**) display of differences between groups, higher (green with black background) and lower (green) functional connectivity in controls compared to patients. L, language network; S, salience network; D, default mode network.

**Table 1 brainsci-11-00356-t001:** Patients’ characteristics.

	Clinical Status	Gender	Age	Handedness	Nature of Initial Injury	Time Elapsed Since Initial Injury	CRS-R Score before MRI (Subscores)
P1	UWS	M	32	R	TBI	12 y 7 m	6 (1 1 1 1 0 2)
P2	MCS	F	63	R	Hemorrhagic stroke	1 y 4 m	10 (2 3 2 1 0 2)
P3	MCS	M	23	L	TBI	3 y 2 m	12 (3 2 4 1 0 2)
P4	MCS	F	22	R	TBI	4 y 2 m	5 (1 1 0 1 0 2)
P5	MCS	F	47	R	Hemorrhagic stroke	2 y 4 m	5 (1 1 0 1 0 2)
P6	UWS	M	42	R	TBI	4 y 11 m	9 (2 1 2 2 0 2)
P7	MCS	M	26	R	TBI with cardiopulmonary arrest	4 y 9 m	10 (2 3 2 1 0 2)
P8	UWS	F	36	R	TBI	7 y 4 m	9 (2 1 2 2 0 2)
P9	MCS	F	50	R	TBI	1 y 4 m	12 (3 3 3 1 0 2)
P10	MCS	M	61	R	TBI	10 m	11 (2 2 3 2 0 2)
P11	MCS	M	20	R	TBI with cardiopulmonary arrest	2 y 1 m	13 (3 3 3 2 0 2)
P12	MCS	F	34	R	Hemorrhagic stroke	5 y 6 m	9 (2 2 2 1 0 2)
P13	UWS	M	45	R	Cardiopulmonary arrest (myocardial infarction)	6 y	6 (1 0 2 1 0 2)

UWS, unresponsive wakefulness syndrome; MCS, minimally conscious state; M, male; F, female; R, right; L, left; TBI, traumatic brain injury; y, year; m, month. The Coma Recovery Scale—Revised (CRS-R) subscores correspond respectively to the auditory, visual, motor, oromotor/verbal, communication and arousal scales.

## Data Availability

The data presented in this study are available on request from the corresponding author. The data are not publicly available due to legal issues.

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
