# Peer review of "Disrupted Pallido-Thalamo-Cortical Functional Connectivity in Chronic Disorders of Consciousness"

_brainsci, 2021, doi:10.3390/brainsci11030356_

Round 1

Reviewer 1 Report

This study evaluated with resting state fMRI the functional connectivity between the pallidum, the thalamus, and several cortical networks in 13 patients with DOC. The author found a disrupted pallido-thalamo-cortical connectivity in patients with DOC. As a general comment, the study is interesting and its findings are in line with the current literature about the mesocircuit model of the DOC pathophysiology. I have the following comments to further improve the manuscript.

Introduction

First line. You may delete the word “permanent”, since recent guidelines suggest that the term “permanent” be replaced by “chronic” to indicate the stability of a DOC (Giacino et al., Neurology 2018).

It is not necessary to use the first capital letters for unresponsive wakefulness syndrome (as they are not used for minimally conscious state).

The mesocircuit model should be introduced with more details.

Methods

It not clear if patients and controls were matched for age and sex. Please, specify.

Two patients with MCS had a low CRS-R score (i.e., 5 points), while one UWS patient had a CRS-R score of 9 that, usually, it is not consistent with a condition of unresponsiveness. Please, add in the table the CRS-R subscores.

Results

P values should be added for all comparisons, for example in a table.

It should be interesting to compare data between patients with UWS and MCS.

Discussions

The authors should add a paragraph with the study’s limitations.

Author Response

We would like to warmly thank the Reviewer for his constructive comments and suggestions which, in our opinion, allowed us to improve the manuscript. Our responses and modifications are listed in the attached file "Reply_Reviewer1", and highlighted in the revised manuscript by using the “Track Changes” function.

Anna Sontheimer

Reviewer 2 Report

In the present study Sontheimer and colleagues employ fMRI recordings in 19 healthy subjects (HS) and 13 chronic patients affected by disorder of consciousness (DOC) - including Unresponsive Wakefulness Syndrome patients (UWS) and patients in Minimally Conscious State (MCS) - to study the functional connectivity of pallidum with thalamic and cortical structures. In line with previous studies (mentioned in the introduction) they found that, with respect to HS, in DOC patients the thalamo-cortical connectivity is abolished as well as the connectivity between thalamus and pallidum. They also found that the connectivity of pallidum with default mode and with salience networks are abolished. Although not very novel per se – many other similar studies are cited by the authors in the introduction – this work reports interesting results on a very specific circuit that have never been studied before and that seems to be key for loss and recovery of consciousness. Indeed, these results perfectly match with the existing literature on the mesocircuit hypothesis.

Overall, this is an interesting manuscript and the reported results can be useful for both the clinical and neuroscientific community. Nonetheless, I have some major and minor concerns that need to be addressed in order to guarantee future reproducibility of the study and proper interpretation of the results. I have detailed my comments below:

1) The patients included in this study are miscellaneous in terms of clinical status (UWS/MCS) and etiology (Table 1). However, in the results section they are all considered as pertaining to the same group of patients (i.e. DOC). I would suggest performing the same analyses performed in the manuscript on DOC, separately on the two groups (UWS and MCS). If the statistical power is not enough to run this kind of analysis the authors should in principle add more subjects to the study. If not possible, this limitation should be clearly mentioned in the results and largely discussed in the manuscripts.   

2) The authors did not report clearly the area(s) affected by the brain damages. This is an important information that must be mentioned somewhere in the text to guarantee reproducibility of the results (perhaps in a table). In particular, the author should indicate whether areas directly adjacent and/or connected to the pallidum are involved in the damage (e.g., pallidum is very often involved in hemorrhagic/ischemic injuries). Finally (optional), something like a correlative analysis between structural and functional damage can be very useful for the interpretation of the results.    

3) “Due to extensive cerebral lesions altering anatomical data processing, functional data were spatially normalized into the MNI space using an EPI template, and resampled at 2 x 2 x 2 mm 3 . The anatomical data were spatially normalized using the same parameters.”
Also in light of point 2 (see above), methodologically, this is the most critical step. I understand that due to lesions, a normalization is required. However, normalization is not well described methodologically and the authors did not discuss the consequences of such a normalization. For instance, it is not clear whether a cost-function-mask was applied for normalization, or whether linear or non-linear steps were implemented. Finally, the authors should indicate whether normalization results were visually checked or not and should explain why normalized maps were not smoothed.

4) “Outliers were detected using ART (http://www.nitrc.org/projects/artifact_detect) and de- 99 fined as volumes with a framewise displacement > 2 mm and 0.035 rad, or with signal intensity changes > 4 times the standard deviations”
Do the authors refer to the averaged x,y,z rotational (mm) and translational (degree) metrics or did they use the framewise displacement metric as reported in Power et al., 2012 (the sum of the absolute values of the derivatives of the six realignment parameters)? In the second case, the cut-off is not the usual cut-off of 0.5/0.25 mm. If they refer to translational and realignment metrics, they should avoid using the term "framewise displacement", which might be misleading. I will also report degrees rather than radians.

5) “a new TPM has been in-house-developed to ensure that the corresponding voxels are not classified as white matter and are not excluded from the functional connectivity analysis (Figure 1). Artificial values were settled to 1 on the grey matter probability map and 0 on the white matter probability map.”
A better definition of the map is advisable. Authors should explain how subcortical regions were identified and implemented in the TPM map (e.g., defined from a different atlas in the same TPM space, from a normative dataset, from an atlas in a different space and then normalized to the TPM, etc). More in general, this step is quite unclear and would need a substantial rephrasing to make it reproducible. Moreover, since thalamus and pallidum are critical regions for this study, would it not be preferable to use different validated tools, such as FreeSurfer, for subcortical segmentation?

6) “Since the signal of interest is considered to belong to the grey matter, the white matter, cerebrospinal fluid, realignment and scrubbing parameters were used as confounders for nuisance regression, with linear detrending. A band-pass filter [0.008 Hz - 0.09 Hz] was applied.”
These sentences refer to processing steps and should be moved from the result paragraph

Author Response

We would like to warmly thank the Reviewer for his constructive comments and suggestions which, in our opinion, allowed us to improve the manuscript. Our responses and modifications are listed in the attached file "Reply_Reviewer2", and highlighted in the revised manuscript by using the “Track Changes” function.

Anna Sontheimer

Round 2

Reviewer 1 Report

The authors responded appropriately to my comments.

Author Response

Thank you for you Review report, we are pleased to have responded appropriately to your comments.

Reviewer 2 Report

The authors addressed all the issues I've reported in the previous round of Review. I suggest to accept the paper for the publication.

Author Response

(The authors gave the same response as above.)
